# Effects of COVID-19 Lockdown on Physical Activity, Sedentary Behavior, and Satisfaction with Life in Qatar: A Preliminary Study

**DOI:** 10.3390/ijerph18063093

**Published:** 2021-03-17

**Authors:** Souhail Hermassi, Maha Sellami, Ahmad Salman, Abdulla S. Al-Mohannadi, El Ghali Bouhafs, Lawrence D. Hayes, René Schwesig

**Affiliations:** 1Physical Education Department, College of Education, Qatar University, Doha 2713, Qatar; msellami@qu.edu.qa (M.S.); asalman@qu.edu.qa (A.S.); 2World Innovation Summit for Health (WISH), Qatar Foundation, Doha 5825, Qatar; absalmohannadi@qf.org.qa; 3Aspetar Orthopaedic and Sports Medicine Hospital, Doha 29222, Qatar; 4Department of Sports Science, Martin-Luther-University Halle-Wittenberg, 06120 Halle, Germany; bouhafs.elghali@gmail.com; 5School of Health and Life Sciences, University of the West of Scotland, Glasgow G72 0LH, UK; Lawrence.Hayes@uws.ac.uk; 6Department of Orthopaedic and Trauma Surgery, Martin-Luther-University Halle-Wittenberg, 06120 Halle, Germany; rene.schwesig@uk-halle.de

**Keywords:** SARS-CoV-2, lockdown, public health, physical activity, sedentary behavior, home confinement, lifestyle and contentment

## Abstract

This study examined the effects of home confinement on physical activity (PA) and life satisfaction during the COVID-19 outbreak in Qatar. A total of 1144 subjects participated (male: n = 588; female: n = 556; age: 33.1 ± 11.1 years; mass: 76.1 ± 16.4 kg; height: 1.70 ± 0.11 m; body mass index (BMI): 26.1 ± 4.44  kg/m^2^). Online survey questions considered “before” and “during” confinement. Confinement reduced all PA intensities (η_p_^2^ = 0.27–0.67, *p* < 0.001) and increased daily sitting time from 3.57 ± 1.47 to 6.32 ± 1.33 h per weekday (η_p_^2^ = 0.67, *p* < 0.001). The largest reduction was detected for the sum parameter all physical activity (minutes per week, η_p_^2^ = 0.67, *p* < 0.001; MET (metabolic equivalent of task)-minutes/week, η_p_^2^ = 0.69, *p* < 0.001). Life satisfaction decreased, with the score for “I am satisfied with my life” (η_p_^2^ = 0.76, *p* < 0.001) decreasing from 28.1 ± 4.81 to 14.2 ± 6.41 arbitrary units (AU). Concerning life satisfaction, the largest change was detected for the statement “the conditions of my life are excellent” (d_male_ = 7.93). For all parameters, time effects were indicative of large negative effects in both genders. In terms of magnitude, the difference between gender was greatest for the parameter “the conditions of my life are excellent” (difference between groups, d = 4.84). In conclusion, COVID-19 confinement decreased PA, increased sitting time, and decreased life satisfaction in Qatar. These precautionary findings explicate the risk of psychosocial impairment and the potential physical harm of reducing physical activity during early COVID-19 confinement in 2020.

## 1. Introduction

The World Health Organization (WHO) declared a pandemic due to COVID-19 on the 11 March 2020. Subsequently, the WHO recommended adopting several protective, behavioral, and non-pharmacological actions such as avoidance of physical contact, handshakes, closing universities and schools, banning social and large gatherings, and implementing self-isolation, social/physical distancing, confinement, and quarantine.

In the face of the ongoing pandemic, public health authorities and governments have enforced increasingly restrictive recommendations, including self-isolation, quarantine and even lockdowns of entire communities and territories [1,2]. While these restrictions help curb infection rate, such limitations may negatively affect participation in daily activities, physical activity (PA), traveling, and access to many forms of unstructured/structured exercise (e.g., closed gyms, no group gatherings, increased social/physical distancing) [1,2]. Numerous communities implemented curfews, limiting time for outdoor activities, or banned outdoor activities entirely. Such restrictions inflict a significant burden on population health by possibly compromising physical fitness, which is known to positively influence the ability to fight infections and reduce more severe immunologic and cardiopulmonary complications [3,4,5].

Regular PA induces cardioprotective physiological adaptations such as increased stroke volume, cardiac output, lowered resting heart rate and blood pressure, which improve individuals’ health-related quality of life [6]. Prolonged homestay could increase behaviors that lead to physical inactivity and increase stress, which in turn, could increase the risk of several chronic health conditions that adversely affect health-related quality of life [6]. For example, limiting daily activity has been found to significantly impair health-related quality of life among severe acute respiratory syndrome (SARs) patients in Hong Kong [7].

The mandated restrictions concerning engagement in outdoor activities, including regular practice of exercise and physical activity in the time of the COVID-19 outbreak, are reducing exercising and increasing sedentary behavior, which can consequently contribute to anxiety, depression and common chronic health diseases [2,8,9,10]. In addition, restrictions likely reduced PA frequency and duration among typically active people who were unable to access gyms and health clubs and among those who achieve sufficient levels of PA incidentally, through walking or cycling to work or study [11,12]. It has been previously reported that COVID-19 confinement produced negative psychological effects, including post-traumatic stress, confusion, or anger [13,14]. During confinement, most individuals are living in an unprecedented situation of unknown duration, being exposed to anxiety, fear, depression, or sleep disruption [15,16].

It is important to note that both governmental policies and public behaviors are likely to vary and change over time depending on the situation of the COVID-19 pandemic. Based on preliminary responses (~1000 participants), the present manuscript aimed to provide insight into the effect of home confinement on PA, sedentary behavior, and life satisfaction. Results from this study may provide guidance related to PA promotion and quality of life improvements, with the ultimate goal of establishing support programs for individuals living during the pandemic. We hypothesized that COVID-19 confinement would, independent of gender, (a) negatively impact PA participation, (b) negatively impact sedentary behavior, and (c) negatively impact life satisfaction. Additionally, we expect this to negatively impact of general population of Qatar state, especially amongst those who reduced their PA levels and increased their sedentary behavior.

## 2. Materials and Methods

An important criterion for selecting questionnaires was the proof and evidence of validity and reliability [17,18]. In this cross-sectional study using a convenience sample, we communicate preliminary findings of the first 1144 responses to an international online survey based on the International Physical Activity Questionnaire Short Form (IPAQ-SF) and Satisfaction with Life Questionnaire (SLQ), which were opened on 22 July 2020, tested by the project’s steering group for one week and spread on 29 July 2020. During data collection in July 2020, the government of Qatar entered its second stage of the normalization plan and adopted a flexible lockdown. The IPAQ and SLQ were administered in English and Arabic. The survey included 22 questions on gender, demographic information (e.g., age, body mass and height), athlete (e.g., defined as a person who competes in one or more activities that involves physical strength, speed, and/or endurance), smoking status, health status (e.g., anxiety or depression, diabetes, cardiovascular disease and pulmonary disease, motor problem), PA (e.g., vigorous, moderate and walking activity) and SLQ (e.g., life conditions, satisfaction with life, and important things for life). Questions were to be answered directly in sequence regarding “before” and “during” confinement conditions [2]. All measures were collected on the same day to avoid order or recall bias in the study, considering the continually evolving situation of the pandemic. Once the deadline for admitting surveys had passed, contradictory responses (incongruence between data), and repeated responses (≥2 submissions with identical responses in a short period), were removed from the database. The study was completed in accordance with the Declaration of Helsinki. The present study’s protocol was approved by the university’s institutional review board (QU-IRB 1350EA-2020).

### 2.1. Sample Size

The sample size was calculated according to the following predictive equation [19].


N = ((Zα/2 2 p q))/Δ2(1)


N: number of needed participants,

Zα/2: two-tailed normal deviate for type 1 error,

p: change in % from “before” to “during” confinement,

q: equal to “1 − p” and Δ: accuracy; where “n” was the number of needed participants,

“Zα/2” was the two-tailed normal deviate for type 1 error (Zα/2 = 1.96 for 95% level of significance),

“q” was equal to “1 − p”,

“Δ” was the accuracy (=3%), and

“p” was the percentage of change in social participation from “before” to “during” confinement period.

Comparable to Ammar et al. [2], the “p” was chosen from a study [20]. Zhang and Ma [20] examined the immediate effects of the COVID-19 pandemic on mental health and quality of life. Based on these findings, it appeared that 57.8% (*p* = 0.578) of subjects experienced an increase in shared feelings with family members [20]. Consequently, the calculated sample size was n = 1041. Assuming a dropout rate of 20% (n = 208), we aimed to recruit 1250 participants.

### 2.2. Survey Development, Promotion, and Distribution

A steering group of academics and scientists (human science, sport science, and computer science) conceived the electronic survey at the University of Qatar (principal investigator). The survey was subsequently evaluated and amended by >30 colleagues and experts. The survey was uploaded and shared via the Google^TM^ platform. The consortium distributed the electronic survey uniform resource locator (URL) via various methods: e-mails, shared on the consortium’s faculties official pages, ResearchGate™, LinkedIn™, and other social media platforms such as Facebook™, Twitter™, and WhatsApp™.

The general public assisted in dissemination through the promotion of the survey within their networks. In total, the URL of the online survey was sent to 1250 potential participants, of which 1144 returned completed questionnaires and were included in the analysis (participation rate of 92%). An introductory page described the survey background and aims, the consortium, ethics information, and the option to choose one of two available languages (English and Arabic). Inclusion criterion was that participants were aged ≥18 years and in good health (no pain and diagnosis at the time of examination). Exclusion criteria included a positive COVID-19 test or existence of cognitive decline.

### 2.3. Data Privacy/Security

Participants were assured all data would be used exclusively for research purposes. Participants’ responses were anonymous and confidential according to Google’s privacy policy (https://policies.google.com/privacy?hl=en, accessed on 22 July 2020). Participants were unable to provide names or contact information to ensure anonymity. Participants were free to withdraw and leave the questionnaire at any time before submission of responses. If they did so, responses would not be saved. Responses were saved once the “submit” button was pressed. By completing the survey, participants acknowledged the approval form and consented to voluntarily participate in this anonymous study. Participants were instructed to be honest with their responses.

### 2.4. International Physical Activity Questionnaire Short Form (IPAQ-SF)

According to the official IPAQ-SF recommendations, data are summed within each item (i.e., vigorous intensity, moderate intensity, and walking) to estimate the total amount of time engaged in PA per week [17,18]. Weekly PA (MET-min·week^−1^) was calculated by summing products of each PA item by a MET value specific to each category of PA. We assigned two sets of MET values. The first was the original values based on official IPAQ procedures for young and middle-aged adults (18–65 years old): vigorous PA = 8.0 METs, moderate PA = 4.0 METs and walking = 3.3 METs. Additionally, we added the total PA (sum of performed vigorous, moderate and walking activity) as a fourth item and sitting time as the fifth item of sedentary behavior assessed using the question: “Since self-isolating, how much time have you spent sitting daily?”

### 2.5. Satisfaction with Life Questionnaire (SLQ)

The Satisfaction of Life Questionnaire (SLQ) is a crisis-oriented questionnaire to assess satisfaction with the respondent’s life before and during the confinement period. The SLQ is based on a 5-item scale designed to measure global cognitive judgments of one’s life satisfaction. Participants indicate how much they agree or disagree with each of the 5 items using a 7-point scale that ranges from 7 strongly agree to 1 strongly disagree. Though scoring should be kept continuous (sum up scores on each item), here are some cut-offs to be used as benchmarks [21]. Using the 1–7 scale below, participants indicated their agreement with each of the five items (Strongly agree = 7; Agree = 6; Slightly agree = 5; Neither agree nor disagree = 4; Slightly disagree = 3; Disagree = 2; Strongly disagree = 1). The total score of this questionnaire is comprised of the sum of scores from individual items. The total score for the SLQ ranges from 5 to 35, with lower scores corresponding to extreme dissatisfaction and higher scores with extreme satisfaction.

### 2.6. Statistical Analyses

All statistical analyses were performed using SPSS version 25.0 for Windows (SPSS Inc., IBM, Armonk, NY, USA). Means and standard deviations of dependent variables were calculated across participants. The chi square test was used to investigate gender differences for dichotomously or ordinally scaled data. Before analyses, all variables were tested for normal distribution (Shapiro–Wilk Test). Differences between males and females and between sessions (before vs. during confinement) were tested using a two-factor (gender, time), univariate general linear model. Within this analysis of variance (ANOVA) *p*-values and partial eta-squared (η_p_^2^) were calculated using the Greenhouse-Geisser Test [22]. In this context, we reported group (gender), time (before, during confinement) and interaction (gender x time) effects.

The effect size (the mean difference between scores divided by the pooled standard deviation) was also calculated for each parameter [23]. A positive effect size means an improvement of performance and a negative value indicates a decrease in performance. Percentage changes were calculated as ((before value − during value)/before value) × 100. The interpretation of effect sizes is based on Cohen’s thresholds for small effects (d < 0.5), moderate effects (d ≥ 0.5) and large effects (d > 0.8) [24].

Differences between means (time effect) were considered as being meaningful if *p* < 0.001, η_p_^2^ > 0.10 and the effect size (d) was ≥ 0.8 [25]. Based on the number of parameters/tests and after applying a Bonferroni correction, we adjusted the α error level for both types of parameters (PA: 0.05/13 = 0.004; satisfaction of life: 0.05/6 = 0.008). In the interest of a uniform approach and conservative assessment of the effects, we defined an alpha error level of *p* < 0.001.

## 3. Results

### 3.1. Sample Description

1144 (female: n = 588; male: n = 556) participants were recruited in Qatar (Table 1).

Significant differences were observed between males and females for height (*p* < 0.001, η_p_^2^ = 0.53) and body mass (*p* < 0.001, η_p_^2^ = 0.34) but not for age (*p* < 0.001, η_p_^2^ = 0.07). BMI was the anthropometric parameter with the smallest difference (*p* < 0.001, η_p_^2^ = 0.04) between males and females.

We found a gender effect (Chi-Square = 111; *p* < 0.001), with the proportion of athletes amongst women higher (45%; 250/556) than among men (16%; 96/588). An effect of gender was observed for number of smokers (Chi-Square = 334.2; *p* < 0.001). The percentage of smokers was different between women (0%) and men (46%; 270/588). Most subjects (76%; 878/1144) reported no health problems (Table 2). Anxiety or depression was the most prevalent health problem reported among the sample (14%; 158/1144).

### 3.2. International Physical Activity Questionnaire Short Form (IPAQ-SF)—Testing of Hypotheses 1 and 2

The time effects for the PA parameters (vigorous or moderate physical activities and walking) and sedentary behavior (measured by sitting time) ranged from η_p_^2^ = 0.27 (vigorous minutes per week) to η_p_^2^ = 0.67 (sitting; hours per weekday; Table 3). The paired effect sizes (d) considering before vs. during confinement were similar in both sexes and ranged from d = 0.58 (female, vigorous physical activities/minutes/week) to d = 2.17 (male, sitting/hours per weekday). The greatest difference between male (d = 1.97) and female (d = 1.30) was observed in moderate PA (MET-minutes/week). Significant gender effects (Table 3) were observed for 46% (6/13) of the investigated parameters. No interaction effects (gender x time) were observed.

### 3.3. Satisfaction with Life Questionnaire (SLQ)—Testing of Hypothesis 3

The comparison of “satisfaction with life” parameters (Table 4) revealed an interaction (time × gender) effect (η_p_^2^ = 0.14) for the statement “So far I have gotten the important things I want in life”. The reduction was smaller in females (absolute difference = 13.4, d = 3.89) than in males (absolute difference = 17.5, d = 5.15).

The time effects reached the *p* < 0.05 level in all parameters and ranged from η_p_^2^ = 0.76 (I am satisfied with my life. Figure 1) to η_p_^2^ = 0.91 (The conditions of my life are excellent. Figure 2). The results concerning “In most ways my life is close to my ideal” (time effect: η_p_^2^ = 0.86; Figure 3) and “If I could live my life over, I would change almost nothing” (time effect: η_p_^2^ = 0.89) were similar. Regarding these two questions, effect size in the subgroups ranged from d = 3.12 to d = 4.94 (Table 4; Figure 3).

The total satisfaction with life demonstrated the greatest reduction in both males (d = 9.02; from 30.8 ± 2.00 to 13.8 ± 1.77) and females (d = 7.61; from 29.4 ± 1.85 to 12.7 ± 2.54).

## 4. Discussion

To attenuate COVID-19 spread, policymakers have implemented restrictive measures in many countries. There are little data on the precise impact of this crisis on population mental health, whether in the short, medium or long term. If links between COVID-19 and mental and behavioral health symptoms are detected, they remain to be validated by large scale cross-sectional or longitudinal studies. During times of crisis, the implementation of severe restrictions and strategies to control and mitigate the spread of disease may have short term or long-term implications on mental health and physical activity. While these restrictions may be intended to protect the publics’ physical health, it may be detrimental to their mental health and well-being. With this in mind, understanding the psychosocial implications of home isolation would provide data on the impact of such health measures on lifestyle (especially physical activity, physical and mental health).

The present study aimed to elucidate the effect of home confinement on PA and life satisfaction based on data extracted from the first thousand responses in Qatar. Home confinement by COVID-19 caused a significant decrease in walking per week (assumption of hypothesis 1) and an increase in sedentary behavior measured by sitting time (assumption of hypothesis 2), which reflects PA in daily life. In addition, PA at vigorous and moderate intensity was decreased by COVID-19 home confinement. Similarly, the total score for life satisfaction decreased by ~56% (male: 55%; female: 57%). Due to the negative impact upon social participation and life satisfaction (assumption of hypothesis 3), data presented here support the conclusion that there is significant risk of psychosocial strain during home confinement conditions caused by the pandemic.

### 4.1. Impact of COVID 19 on Physical Activity

The positive health benefits of PA and sedentary behavior in daily life are well established and there is clear evidence linking physical inactivity to non-communicable diseases [26]. In addition, many governmental agencies have developed PA guidelines not only as a preventive strategy for chronic diseases, but also for psychological benefits [27,28,29]. Recent multicenter studies showed that COVID-19 home confinement increased the numbers of physically inactive individuals (+15%) [2,11]. In this study, the number of walking days for at least 10 min per week was decreased of ~35% in both genders. Concomitantly, time per walk decreased of 46% in both genders. Therefore, the energy expenditure of walking per week was decreased from 449 ± 261 MET-minutes/week to 141 ± 87.0 MET-minutes/week for women and from 528 ± 271 MET-minutes/week to 215 ± 120 MET-minutes/week in men. It is thought that walking PA is part of the habitual activities of daily life. Thus, the decreased walking volume in our subjects may reflect an inactive lifestyle. Ammar et al. [30] reported after an international online survey was launched in April 2020 that the number of days/week of walking decreased by 35% during home confinement (t = 15.80, *p* < 0.001, d = 0.68). Likewise, the same study reported that the number of minutes/day of walking decreased by 34% during home confinement (t = 9.34, *p* < 0.001, d = 0.39). Additionally, MET values of walking were 43% lower during home confinement (t = 9.03, *p* < 0.001, d = 0.36). It is important to mention here that the studies were all developed during the cool weather, which make comparison easier between populations from North Africa, Europe and Asia.

Results of PA at vigorous intensities were consistent with those of previous studies in Italian athletes [31]. The frequency of vigorous intensity PA during home confinement decreased from 40% to 23% in women and from 32% to 30% in men, respectively. Giustino et al. [31] reported energy expenditures in vigorous intensity were decreased from 520 ± 372 MET-minutes/week to 238 ± 205 MET-minutes/week in women and from 663 ± 320 MET-minutes/week to 323 ± 187 MET-minutes/week in men by home confinement. However, Ammar et al. [30] indicate that the number of days/week and minutes/day of vigorous intensity PA during confinement decreased 23% (t = 7.75, *p* < 0.001, d = 0.37) and 33% (t = 9.75 *p* < 0.001, d = 0.54), respectively. Moreover, vigorous intensity METs were 37% lower during home confinement (t = 6.68, *p* < 0.001, d = 0.32) compared to before. In addition, in a survey of PA during lockdown in the Canadian adult population [12], vigorous PA among highly active people (Moderate to Vigorous Physical Activity (MVPA) of 302 ± 186 min per week) did not decreased despite the lockdown. Their vigorous PA consisted of individual outdoor exercise such as walking, running and cycling. In contrast, amounts of vigorous and moderate intensity PA in this study decreased. The difference between these results could be the difference between individual PA for recreation and health promotion. However, Ammar et al. [30] found that moderate intensity PA, in days per week, decreased by 24% during home confinement (t = 7.89, *p* < 0.001, d = 0.40). Likewise, minutes/day of moderate intensity PA decreased 33% during home confinement (t = 7.85, *p* < 0.001, d = 0.34). As a result, moderate intensity METs were 35% lower during confinement (t = 5.24, *p* < 0.001, d = 0.20).

People undertaking individual PA for recreation and health promotion may continue the same exercise during lockdown because they are outdoor activities such as walking, running, and cycling. Conversely, individuals involved in indoor sports obtained their MVPA almost exclusively through handball training in training facilities that closed during restrictions. Most athletes who usually complete PA in training facilities have been forced to train at home or in their own backyards [2,12,31]. Thus, some of the athletes in this study would no longer be able to gather in training facilities for training and matches and likely trained individually in their own homes and/or backyards. This may have profound implications for the athletic participants within this study as the inability to maintain intensity and volume of habitual training would result in reduced physical performance [32,33,34]. Moreover, the transition back into sport-specific practices would result in a high risk of injury [35].

Sitting time would be considered an indicator of how much time is spent in the home, and we report herein that sitting time on weekdays increased almost twofold in both genders. Previous studies have reported healthy populations increased time watching television, social networking using smart phones, and video gaming during COVID-19 home confinement [30,36]. Ammar et al. [30] also reported the number of hours/day sitting increased 29% during home confinement. Thus, the impact of home confinement by COVID-19 on physical inactivity in daily life in this study is commensurate with results from earlier investigations.

Statistics from the current sample showed a slightly high percentage of individuals with depression and anxiety problems (anxiety or depression: 14%). Such results were expected, as the current pandemic is a time of uncertainty and concern for many people. This can affect individuals on a physical level, but also on a psychological level. Indeed, in such a context, many people will experience reactions of stress, anxiety and depression. Interestingly, in our study the proportion of smokers (24%), people with diabetes (2%), people with cardiovascular disease and pulmonary disease (3%) and motor problems (5%) was much lower compared to the general population. Through recent decades, Qatar has developed a healthcare strategy for prevention, cure, and care to increase awareness of health and fitness [37] which has led to lower rates of mortality when infected [38].

### 4.2. Impact COVID 19 on Life Satisfaction

COVID-19 induced social isolation negatively impacted mental health. In 1006 Italians under COVID-19 quarantine, prolonged isolation increased depression, unworthiness, alienation, and helplessness [39]. Similarly, adults ceasing work after one month of confinement in China (2020) reported worse health circumstances and mental distress. The present findings revealed a significant interaction effect for the statement “So far I have gotten the important things I want in life” as the reduction was less in females than males. The time effect reached *p* < 0.05 in all parameters (I am satisfied with my life, the conditions of my life are excellent). Ammar et al. [2] showed that the total score for Life Satisfaction decreased by 16% during home confinement. This overall score is a factor of three questions (Q1–Q3), and the decrease from “before” to “during” confinement ranged from −14% to −18%. These negative effects have also been reported in a recent COVID-19 series highlighting that people under quarantine conditions report more symptoms of psychological distress. Furthermore, some symptoms appeared to persist after quarantine has ceased [40]. In China, COVID-19’s resultant social distancing reduced life satisfaction and increased distress [41]. The present findings corroborate these previous reports, substantiating the risk of psychosocial strain during home confinement periods.

Ammar et al. [42] indicated that COVID-19 home confinement had a negative effect on mental health, mood and feelings. In terms of prevalence, more individuals (+13%) reported lower mental wellbeing during, compared to before, home confinement. Mood and feelings responses showed a 45% increase in depressive symptoms, with more people (+10%) showing depressive symptoms during, compared to before, home confinement. The ECLB-COVID-19 survey revealed an increased psychosocial strain triggered by home confinement. To mitigate this risk of poor mental health and to foster an active and healthy confinement lifestyle (AHCL), a crisis-oriented interdisciplinary intervention is urgently needed.

Associations between mental health and quality of life have recently been reported [43]. However, factors such as income, which in turn is relates to factors such as education and health literacy, mediate associations between mental health and quality of life in SARs patients [44,45]. Apart from the above influences, Qi et al. [46] reported stress and health-related quality of life was strongly negatively associated among Chinese adults during the COVID-19 pandemic. Psychological perturbations in SARs infected patients during the quarantine periods appear relatively ubiquitous [47]. Home confinement negatively influences mental wellbeing and emotional status, potentially due to physical inactivity, social isolation, and unemployment [2]. Importantly, sedentary behavior due to COVID-19 negatively impacted people’s quality of life in this study. Therefore, it is critically important to promote people’s health and wellbeing by encouraging them to minimize sitting time and keep physically active to maximize satisfaction with life during the COVID-19 pandemic. Further studies are needed to confirm the association between stress and health-related quality of life during or after the COVID-19 pandemic.

### 4.3. Strengths, Limitations, and Perspective

The strength of this study was the rapid data generation during the pandemic using an anonymous cross-disciplinary survey provided in two languages, widely distributed over several region in Qatar. However, given the preliminary nature of this study, moderating effects of demographics, cultures, and age have not been studied. Regarding the methodological issues, possible limitations could be (i) the cross-sectional design assessing “before” home confinement condition retrospectively, and (ii) the disuse of cookie-based or I.P.-based duplicate protection to exclude duplicates. Our consortium elected to avoid I.P. or cookie safety measures, as we know that during home confinement more than one family member may use the same computer (e.g., same I.P.). Given that home confinement was a sudden measure in most countries, we were obviously unable to develop and spread the survey “before” home confinement, to avoid recall bias. Additionally, different types of PA, divergent professions, presence of children in the home, loss of job, and loss of friends or relatives may be confounding factors on SLQ and perceived stress, which were not accounted for in the present study due to the need for rapid data gathering, and these could be considered in future studies. The compliance regarding the rigid rules of lockdown is also an important predictor.

## 5. Conclusions

Besides stresses inherent to the illness itself, results from the PA COVID-19 survey revealed a deleterious effect of confinement conditions on PA and significantly increased sitting time. These observations have repercussions that could aid development of PA guidelines to maintain health during COVID-19 and subsequent pandemics. Furthermore, these preliminary results confirm the risk of psychosocial strain during early COVID-19 home confinement in 2020. Therefore, to mitigate negative psychosocial effects of home confinement, implementation of national and international initiatives focused on social inclusion is strongly suggested. Increased psychosocial strain triggered by enforced home confinement should encourage stakeholders and policymakers to implement a crisis-oriented interdisciplinary intervention to mitigate the negative effects of restrictions and to foster an active and healthy confinement lifestyle (AHCL). Finally, given that present findings are founded on data from a heterogenous population without subsample analysis, further research is warranted to identify subpopulations at greater risk of COVID-19 confinement measures. Identification of such populations would allow for better informed and more targeted mitigation strategies.

## Figures and Tables

**Figure 1 ijerph-18-03093-f001:**
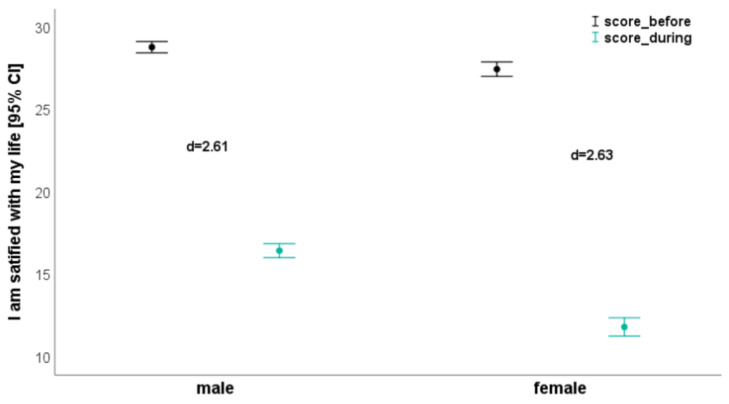
I am satisfied with my life—depending on gender before and during confinement.

**Figure 2 ijerph-18-03093-f002:**
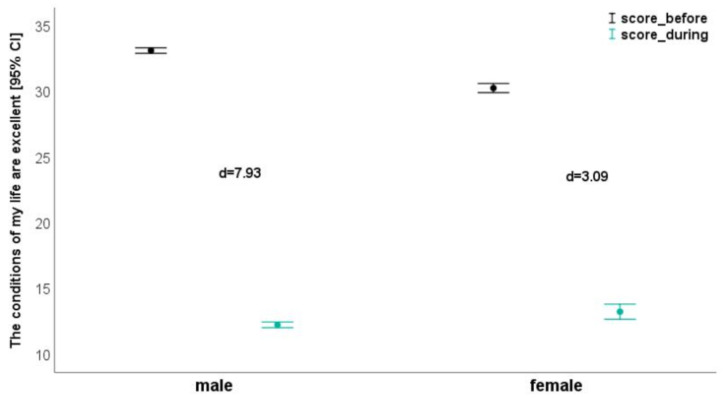
The conditions of my life are excellent—depending on gender before and during confinement.

**Figure 3 ijerph-18-03093-f003:**
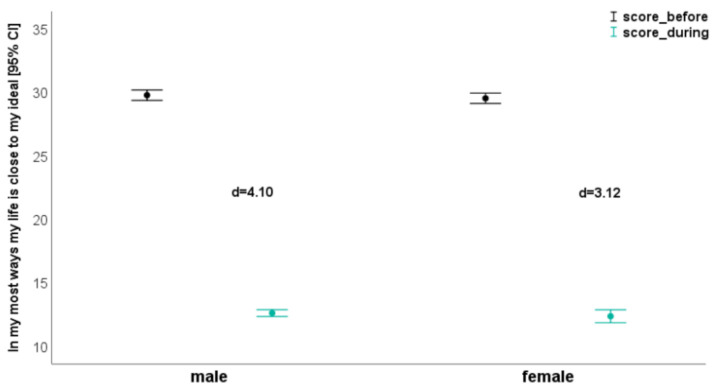
In my most ways, my life is close to my ideal—depending on gender before and during confinement).

**Table 1 ijerph-18-03093-t001:** Demographic and anthropometric characteristics of all participants (n = 1144). Significant effects (criteria of relevance: *p* < 0.05 and η_p_^2^ > 0.10 and d > 0.8) highlighted in bold.

	Total (n = 1144)	Male (n = 588)	Female (n = 556)	*p*	η_p_^2^
Age (yr)	33.1 ± 11.1 (18.0–67.0)	35.9 ± 11.4 (18.0–67.0)	30.0 ± 9.90 (18.0–65.0)	<0.001	0.07
Height (m)	1.70 ± 0.10 (1.40–1.98)	1.78 ± 0.10 (1.55–1.98)	1.62 ± 0.10 (1.40–1.81)	<0.001	0.53
Mass (kg)	76.1 ± 16.4 (28.0–142)	85.4 ± 13.1 (50.0–142)	66.2 ± 13.4 (28.0–140)	<0.001	0.34
BMI (kg/m^2^)	26.1 ± 4.44 (11.7–49.1)	27.0 ± 4.15 (16.4–49.1)	25.1 ± 4.54 (11.7–44.7)	<0.001	0.04

Results reported as mean ± standard deviation (range). BMI = body mass index.

**Table 2 ijerph-18-03093-t002:** Description of the sample (n = 1144; female: n = 556, male: n = 588) regarding demography, anthropometry and health status depending on gender.

Variables	Categories	Male n (%)	Female n (%)	Total n	Chi^2^ (*p*)
Age (yr)	18–35	308 (43)	409 (57)	717	55.7 <0.001
	36–55	255 (65)	138 (35)	393	
	>55	25 (74)	9 (26)	34	
BMI (kg/m^2^)	<18.5	5 (33)	10 (67)	15	43.6 <0.001
	18.5–24.9	206 (41)	297 (59)	503	
	25–29.9	259 (60)	175 (40)	434	
	30 or greater	118 (62)	74 (38)	192	
Athlete	Yes	96 (28)	250 (72)	346	111.1 <0.001
	No	492 (62)	306 (38)	798	
Smoker	Yes	270 (100)	0 (0)	270	334.2 <0.001
	No	318 (36)	556 (64)	874	
Health status	None of the above	452 (52)	426 (68)	878	93.8 <0.001
	Anxiety or depression	117 (74)	41 (26)	158	
	Diabetes	0 (0)	19 (100)	19	
	Cardiovascular disease and pulmonary disease	0 (0)	32 (100)	32	
	Motor problem	19 (33)	38 (67)	57	

**Table 3 ijerph-18-03093-t003:** Comparison of physical activity parameters and sedentary behavior (sitting time) between males and females before and during confinement. Values are given as mean ± SD. Significant effects (criteria of relevance: *p* < 0.05 and η_p_^2^ > 0.10 and d > 0.8) highlighted in bold.

	Male (n = 588)			Female (n = 556)			Variance Analysis/Effects *p* (η_p_^2^)		
	Before	During	d	Before	During	d	Gender	Time	Gender × Time
Vigorous physical activities									
Days/week (d)	2.09 ± 0.81	1.43 ± 0.56	**0.96**	2.44 ± 1.00	1.46 ± 0.82	**1.08**	<0.001 (0.03)	**<0.001 (0.35)**	<0.001 (0.02)
Minutes/week (min)	39.7 ± 11.3	28.0 ± 10.5	**1.07**	25.7 ± 12.1	19.7 ± 8.48	0.58	**<0.001 (0.34)**	**<0.001 (0.27)**	<0.001 (0.04)
MET-minutes/week	663 ± 320	323 ± 187	**1.34**	520 ± 372	238 ± 205	**0.98**	<0.001 (0.07)	**<0.001 (0.39)**	0.012 (0.01)
Moderate physical activities									
Days/week (d)	2.42 ± 0.66	1.56 ± 0.54	**0.99**	2.16 ± 0.82	1.46 ± 0.63	**0.97**	<0.001 (0.03)	**<0.001 (0.41)**	0.003 (0.01)
Minutes/week (min)	40.4 ± 11.2	24.2 ± 9.05	**1.60**	35.3 ± 14.3	21.0 ± 9.51	**1.20**	<0.001 (0.06)	**<0.001 (0.50)**	0.035 (0.00)
MET-minutes/week	394 ± 166	152 ± 79.7	**1.97**	315 ± 188	130 ± 96.3	**1.30**	<0.001 (0.06)	**<0.001 (0.54)**	<0.001 (0.02)
Walking									
Days/walk for at least 10 min (d)	4.28 ± 1.47	2.74 ± 0.74	**1.39**	3.06 ± 1.09	2.07 ± 0.92	**0.99**	**<0.001 (0.24)**	**<0.001 (0.45)**	<0.001 (0.04)
Minutes per/walking days (min)	37.0 ± 12.5	23.0 ± 8.59	**1.33**	43.5 ± 17.2	20.6 ± 8.73	**1.77**	<0.001 (0.01)	**<0.001 (0.57)**	<0.001 (0.07)
MET-minutes/week	528 ± 271	215 ± 120	**1.60**	449 ± 261	141 ± 87.0	**1.77**	<0.001 (0.06)	**<0.001 (0.58)**	0.767 (0.00)
Sitting									
Hours per weekday (h)	3.64 ± 1.42	6.51 ± 1.22	**2.17**	3.49 ± 1.53	6.12 ± 1.41	**1.79**	<0.001 (0.02)	**<0.001 (0.67)**	0.040 (0.00)
All Physical Activity									
Days/week (d)	2.93 ± 0.57	1.91 ± 0.37	**2.17**	2.55 ± 0.61	1.67 ± 0.53	**1.54**	**<0.001 (0.13)**	**<0.001 (0.66)**	0.001 (0.01)
Minutes/week (min)	117 ± 21.4	75.1 ± 16.9	**2.19**	105 ± 31.3	61.2 ± 16.0	**1.85**	**<0.001 (0.14)**	**<0.001 (0.67)**	0.468 (0.00)
MET-minutes/week	1584 ± 447	689 ± 249	**2.57**	1283 ± 582	509 ± 271	**1.82**	**<0.001 (0.14)**	**<0.001 (0.69)**	<0.001 (0.01)

**Table 4 ijerph-18-03093-t004:** Comparison of “satisfaction with life” parameters depending on sex before and during confinement. Values are given as mean ± SD. Significant effects (criteria of relevance: *p* < 0.05 and η_p_^2^ > 0.10 and d > 0.8) highlighted in bold.

	Male (n = 588)			Female (n = 556)			Variance Analysis/Effects *p* (η_p_^2^)		
	Before	During	d	Before	During	d	Sex	Time	Sex × Time
In most ways my life is close to my ideal									
Score Q1	29.8 ± 5.06	12.6 ± 3.34	**4.10**	29.5 ± 4.85	12.4 ± 6.11	**3.12**	0.230 (0.00)	**<0.001 (0.86)**	0.989 (0.00)
The conditions of my life are excellent.									
Score Q2	33.1 ± 2.63	12.2 ± 2.64	**7.93**	30.3 ± 4.15	13.2 ± 6.92	**3.09**	<0.001 (0.02)	**<0.001 (0.91)**	<0.001 (0.09)
I am satisfied with my life									
Score Q3	28.8 ± 4.26	16.4 ± 5.25	**2.61**	27.5 ± 5.26	11.8 ± 6.67	**2.63**	**<0.001 (0.14)**	**<0.001 (0.76)**	<0.001 (0.04)
So far I have gotten the important things I want in life.									
Score Q4	30.5 ± 4.02	13.0 ± 2.78	**5.15**	28.6 ± 2.86	15.2 ± 4.03	**3.89**	0.241 (0.00)	**<0.001 (0.90)**	**<0.001 (0.14)**
If I could live my life over, I would change almost nothing.									
Score Q5	31.8 ± 3.56	15.0 ± 4.52	**4.94**	31.1 ± 4.73	10.9 ± 5.54	**3.93**	**<0.001 (0.12)**	**<0.001 (0.89)**	<0.001 (0.06)
Total score									
	30.8 ± 2.00	13.8 ± 1.77	**9.02**	29.4 ± 1.85	12.7 ± 2.54	**7.61**	**<0.001 (0.17)**	**<0.001 (0.97)**	0.117 (0.00)

Q = Question.

## Data Availability

The raw data supporting the conclusions of this article will be made available by the authors, without undue reservation.

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
