# Peer review of "Effects of COVID-19 Lockdown on Physical Activity, Sedentary Behavior, and Satisfaction with Life in Qatar: A Preliminary Study"

_ijerph, 2021, doi:10.3390/ijerph18063093_

Round 1

Reviewer 1 Report

Overall, this is a well-designed, implemented, and written manuscript. Below are my comments I would like to see addressed.

Keywords

-Include “physical activity”, “sedentary behavior”

Title

-Add sedentary behavior or sitting time to the title, since this was a primary dependent variable.

Abstract

-Include the p-values in the abstract.

-Line 21-22: delete “volume of”.

-Line 22: provide a unit for sitting time (e.g., minutes, hours, etc.).

-Line 29-30: add that daily sitting time was increased in the second to last sentence.

-Line 30-31: you only indicate about psychosocial duress; also include the potential physical harm of reducing PA and increasing SB.

Introduction

-Line 49-50: remove the hyphen from  “popula-tion” and “influ-ence”. There are many other incorrectly placed hyphens throughout the manuscript. Fix these.

-Line 49: change “positive” to “positively”.

-Lines 52-53: remove the “and’s” to replace with comma.

-You discuss sedentary behavior and physical inactivity; it is also included in your measurements. Therefore, provide sedentary behavior in your aims and hypotheses. Because sedentary behavior is independent of PA, you need to separate the two throughout the paper.

Methods

-Line 85-85: remove “a f”.

-Line 86: remove “SL” from “SLSQL”.

-Line 85: What is the project’s steering group?

-Line 90: Provide examples of the SLQ questionnaire, as you did for the PA questionnaire.

-The IPAQ-SF assesses sedentary behavior. You have included sedentary behavior in your results, but it is not described how it is assessed in the methods. Please include this in the methods.

Results

-Table 1: in the table description, for BMI, you have np^2 = 0.04 but the table says 0.07. Additionally, the p-value for age and BMI are both significant but you state there is no significant difference here. Please change.

-Table 2: in the methods, you describe those chronic disease or orthopedic conditions were excluded. Then, table 2 illustrates there were subjects who had these diseases. Please fix this or explain why.

-Lines 209-210: separate PA and sedentary behavior into two separate parameters (as I have previously indicated).

-Table 3: include “sedentary behavior” into the table description.

Discussion

-Line 278: add “physical health”.

-Line 280: change “motor” to something such as “movement”.

-Lines 293-294: Again, separate PA and sedentary behavior (have two different citations here).

-4.1. Impact of COVID 19 on Physical Activity: I would move the second paragraph that discusses sitting time to the second to last paragraph. That way, you have multiple paragraphs in a row that discuss PA, then the second to last paragraph discusses sitting.

Author Response

We thank the Editor for reviewing our manuscript thoroughly and providing constructive feedback. We believe the quality of our manuscript has certainly improved as a result of these comments. Our responses to the comments of Reviewer 1 and the necessary changes are included in the revised manuscript. We have listed the comments and provided our responses here.

Souhail Hermassi, Ph, D.

Reviewer 2 Report

Dear Editor,

Thank you for the opportunity to review this interesting manuscript. Briefly, the authors examined effects of home confinement on physical activity (PA) and life satisfaction during the COVID-19 outbreak in Qatar and found that COVID-19 confinement decreased PA and life satisfaction in Qatar. In spite of the manuscript subject be updated, there are some concerns that prevent it publication in current form.

Below I describe my impressions/doubts about manuscript:

Major concerns:

  • The authors investigated PA and life satisfaction in independent manner. It would be interesting also to investigate if changes in PA levels impair life satisfaction.
  • The authors did not describe the lockdown characteristics of Qatar during data collection. This is mandatory because several countries adopted a rigid lockdown, while other countries adopted a flexible lockdown.
  • The authors did not describe participants characteristics. For example, I think those people who lost their jobs due to the pandemic would present worst life satisfaction. In this line, those participants who lost friends and relatives due to COVID-19 present worst life satisfaction. What do you think about it?
  • Other important limitation of study is related with lockdown adoption. It is possible that there are participant in sample that adopted a rigid lockdown and other that did not follow lockdown. It can impact the results. What do you think about it?

Minor concerns:

1) Line 19: It is not common to start a sentence with a number. Change to ‘A total of 1144 subjects...’

2) Lines 28/29: Which sex had the lowest result? You need to be more specific.

3) Line 31: Change ‘duresss’ to ‘impairment’.

4) Line 32: Intensity is a very broad word. I suggest changing to a more specific word.

5) Line 36: Or declared pandemic due to COVID-19?

6) Lines 49-51: ‘popula-tion’, ‘influ-ence’, ‘pul-monary’. Remove hyphen.

7) Line 58: ‘SARs’. The author must define in first apparition.

8) Line 61: I think the best term is not physical activity but sedentary behavior.

9) Line 71: Change ‘epidemic’ to ‘pandemic’

10) Line 85: To delete an extra ‘f’ in the end of line.

11) Line 127:  ‘uniform resource locator’ To delete because you defined it in line 123.

12) Line 131: I didn't understand the reason for having a questionnaire in English and Arabic.

13) Line 132: It was not clear why you chose only healthy volunteers. What is the criterion for having good health? Especially for older participants, it is probable that many did not present good health.

14) The authors need to state that was investigated a convenience sample.

15) Line 172: Which normality test did you use?

16) Line 181: Change ‘pre-training’ to ‘before value’

17) Table 1: You stated in line 132 that were selected participants over than 18 years old. However, there are participants with 17 years old. Please clarify.

18) Table 2: What was the definition of athlete used by the authors? This needs to be made clear in the methods.

19) Table 2: Was the person who smoked 1 cigarette treated the same as the person who smoked 20 cigarettes daily?

20) Line 201: You need to describe that you did the Chi-square test in the statistical analysis section.

21) Lines 293-300: If you knew that, why did you do this study?

22) Lines 342-344: Not necessarily, because there are people who do activity for recreation and health promotion at gyms. In addition, in some countries the movement of people has been restricted, hence the need to describe the characteristics of the lockdown in Qatar at the time of the study.

Author Response

(The authors gave the same response as above.)

Reviewer 3 Report

The study is relevant and the sample size is large which is suitable for an epidemiology study. However I do have some concerns regarding the study, which are as follows:

- Only one hypothesis was proposed yet there were several statistical tests performed. Please ensure that there is alignment so that the tests are matched with a hypotheses (results to Introduction connection), and that the plan to investigate each hypothesis is clear (Introduction to Analysis Plan connection)

- Study design should be mentioned in the Methods, which was cross-sectional

- Please explain "the p was chosen from a study", shouldn't the p, by default be .05 or less?

- When and how the participants consented to participated in the study is missing

- Please ensure that all analyses shown in resulted are presented under "Statistical Analysis". For instance, section 2.6 should mention a plan to conduct a chi square test to investigate gender differences

- It's not clear how the Np2 effect size was derived. Generally this is the result of an F test, such as an ANOVA, but no F tests were reported or planned. It is also not mentioned under the Statistical Analysis how Np2 would be derived.

- A lot of statistical information is missing. For instance "No interaction effects (gender x time) were observed" There is no information for the reader on how this was performed. There was also nothing mentioned
in the Analysis plan regarding the interaction. It seems random.

Author Response

We thank the Editor for reviewing our manuscript thoroughly and providing constructive feedback. We believe the quality of our manuscript has certainly improved as a result of these comments. Our responses to the comments of Reviewer 3 and the necessary changes are included in the revised manuscript. We have listed the comments and provided our responses here.

Souhail Hermassi, Ph, D.

Round 2

Reviewer 2 Report

Dear Editor

The authors addressed all my concerns and I think that manuscript can be accepted in current form. Congratulations!

Reviewer 3 Report

The authors mention "life score" in the results for the first time. This is not mentioned previously. I believe it is the "total satisfaction with life" score, but they need to clarify this.